# Automated detection and quantification of two-spotted spider mite life stages using computer vision for high-throughput *in vitro* assays

**Michele S. Wiseman**[1], **Joanna L. Woods**[2], **Carly R. Hartgrave**[3], **Briana J. Richardson**[1], **David H. Gent**[1,2]*

**1** Department of Botany and Plant Pathology, Oregon State University, Corvallis, Oregon, United States of America, **2** Forage Seed and Cereal Research Unit, United States Department of Agriculture, Agricultural Research Service, Corvallis, Oregon, United States of America, **3** Department of Soil Sciences, Oregon State University, Corvallis, Oregon, United States of America

* dave.gent@usda.gov

## Abstract

The two-spotted spider mite (*Tetranychus urticae* Koch) is a globally significant agricultural pest with high reproductive capacity, rapid development, and frequent evolution of miticide resistance. Breeding and selection of resistant host cultivars represent a promising complement to chemical control, but widespread adoption is limited primarily due to the labor-intensive nature of conventional *in vitro* phenotyping methods. Here, we present a high-throughput, semi-automated image analysis pipeline integrating the Blackbird CNC Microscopy Imaging Robot with computer vision models for mite life stage identification. We developed a publicly available dataset of over 1,500 annotated images (nearly 32,000 labeled instances) spanning five biologically relevant classes across 10 host species and >25 cultivars. Three YOLO11-based object detection models (three-, four-, and five-class configurations) were trained and evaluated using real and synthetic data. The three-class model achieved the highest overall performance on the hold out test set (precision = 0.875, recall = 0.871, mAP50 = 0.883), with detection accuracy robust to host background and moderate object densities. Application to miticidal assays demonstrated reliable fecundity estimation but reduced accuracy for mortality assessment due to misclassification of dead mites. In hop cultivar assays, the pipeline detected significant differences in fecundity, aligning with manual counts ($R^2 \geq 0.98$). Performance declined on hosts absent from training data and at densities exceeding ∼80 objects per image, underscoring the need for host-specific fine-tuning and density-aware assay experimental design. By enabling rapid, standardized, and reproducible quantification of mite life stages, this system offers a scalable alternative to manual scoring, particularly for resistance breeding programs targeting antibiosis traits. Our approach addresses major throughput bottlenecks in *T. urticae* phenotyping and establishes a framework

**Data availability statement:** Code: All code used for model training, testing, inference, and figure building is available at: http://github.com/mswiseman/miteVision 1500+ labeled image dataset: Users can download or clone our dataset, or upload their own images or videos for free inference using our Roboflow-hosted models, available at: https://universe.roboflow.com/gent-lab/ Model weights Version 209, 210, and 211 model weights are available for download at: https://zenodo.org/records/16945337 Test sets: The referenced test sets can be downloaded from: https://zenodo.org/records/16945379.

**Funding:** The authors received a Blackbird Imaging Robot through a sub-award from the USDA ARS AI Innovation Fund Cooperative Agreement (No. 0440860). This research was conducted in support of United States Department of Agriculture CRIS project 2072-21000-061-000-D.

**Competing interests:** The authors have declared that no competing interests exist.

for integrating automated imaging into broader pest management and plant breeding pipelines. Dataset, code, and trained models are publicly available to facilitate adoption and extension.

## Introduction

The two-spotted spider mite (TSSM, *Tetranychus urticae* Koch, Acari: Tetranychidae) is a pervasive agricultural pest, feeding on over 1,100 plant species [1]. Symptoms of TSSM feeding can include stippling, discoloration, foliar desiccation, and defoliation, contributing to reduction in both crop quality and yield. Frequent scouting and early control of TSSM are essential for effective pest management; nonetheless, even when employing best practices, control remains challenging due to difficulties in achieving adequate spray coverage, the plethora of possible nearby host reservoirs, the selection of miticide-resistant populations, and the mites' high reproductive rate that facilitates rapid population growth [2].

Further complicating control efforts, the two-spotted spider mite's short life cycle, high reproductive output, and unique haploid-diploid sex determination all contribute to rapid evolution of miticide resistance. As of 2025, TSSM miticide resistance has been documented in 96 active ingredients (Arthropod Pesticide Resistance Database accessed 2025), more than any other arthropod [3]. Under optimal conditions, the TSSM lifecycle can complete in 7 to 12 days and females can lay over 300 eggs in their lifetime [4]. The combination of quick generational turnover and high fecundity rates enables TSSM populations to quickly select for genetic variants resistant to miticides. Adding to this, TSSM reproduces through arrhenotokous parthenogenesis whereby unfertilized eggs develop into haploid males, while fertilized eggs become diploid females [5]. This reproductive strategy allows for increased prevalence of recessive traits, including those that confer resistance to miticides, thereby accelerating the development of resistance to chemical control methods [6]. Together, these intrinsic biological traits pose significant obstacles to long-term chemical control strategies.

Developing cultivars resistant to *T. urticae* offers a promising approach to mitigating the two-spotted spider mite's impact on agriculture. Forms of resistance can be broadly classified as tolerance, antixenosis, and antibiosis [7]; these mechanisms can act independently or synergistically to confer resistance to *T. urticae* and other herbivores across diverse plant species [8]. Current methods for evaluating miticide efficacy and host resistance remain a major bottleneck. Traditional *in vitro* assays assess traits such as fecundity, development, mite mortality, leaf damage, and repellency over time [9–13]. Several studies have examined correlations among these traits, as they are directly linked to outbreak risk; however, this approach is both labor-intensive and time-consuming thus limiting scaling and highlighting the need for higher-throughput methods.

Evidence of host resistance mechanisms has been documented in several crops. In eggplant, Taher et al. (2019) found that TSSM-resistance was correlated with the presence of type VII glandular trichomes in *Solanum macrocarpon* and *S. aethiopicum*, while stellate trichome density showed no association [14].

In wild tomato (*S. pimpinellifolium*), Alba et al. reported that high acylsucrose content and dense type IV glandular trichomes enhanced mite mortality, repellence, and reduced oviposition [15]. In cassava, Chen et al. (2022) demonstrated that resistance involved both physical and biochemical defenses; transgenic overexpression of tannin biosynthesis genes increased tannin levels, conferring TSSM resistance [16]. These diverse mechanisms offer multiple breeding targets and suggest that combining strategies may yield more durable resistance. High-throughput resistance screening approaches may help uncover additional mechanisms and accelerate the identification of resistant germplasm.

Image analysis is increasingly being used to detect and quantify arthropods[17]. Combining automated microscopic imaging with computer vision modeling offers a promising strategy for improving the throughput of *in vitro* TSSM assays. Złotkowska et al. ([18]) recently demonstrated the effective integration of autonomous imaging systems with computer vision models to identify accurately and quantify TSSM leaf damage, as well as enumerate mites, eggs, and feces on *Arabidopsis thaliana* leaves. These technologies can enable semi-automated higher-throughput detection and enumeration of the life stages of TSSMs on independent samples under highly controlled conditions with little human interaction. Further, utilization of this approach may improve reproducibility of bioassay assessments thus enabling more precise phenotyping and efficient development of resistant germplasm.

Our long-term objective is to integrate computer vision–based methods into *in vitro* TSSM assays to enable rapid, standardized, and reproducible assessments of mite life stages on diverse plant genotypes. In the present study, we take a step toward this goal by developing and evaluating a high-throughput image acquisition and analysis pipeline tailored to *in vitro* TSSM assays. Specifically, we (1) developed a publicly available, 1500+ annotated image dataset of TSSM life stages collected across 10 host species and more than 25 cultivars; (2) developed and benchmarked three multi-class object detection models capable of accurately identifying biologically relevant mite life stages; and (3) assessed the implementation and limitations of fecundity and miticide *in vitro* assays using our computer vision models on a commercially available imaging platform. Together, these contributions provide a foundation for semi-automated *in vitro* TSSM phenotyping and support the integration of image-based monitoring into TSSM management and resistance breeding programs.

## Materials and methods

### Data collection and dataset overview

Images used to train and evaluate the models were collected through three primary methods: (1) direct imaging of infested leaves or 0.8 cm leaf disks, (2) imaging of 0.8 cm stickers coated with dislodged mites from a mite brushing machine (Juchheim Laborgeräte GmbH, Germany), and (3) synthetic images generated by segmenting and pasting augmented objects onto diverse backgrounds. Most images were captured on hop or bean leaves —- bean being a preferred host of *T. urticae* [19], and hop offering visual complexity due to its trichomes and lupulin glands, thereby improving model generalizability. The final dataset comprises over 1,300 annotated real images and 200 synthetic images, totaling nearly 32,000 labeled instances across five biologically relevant classes: *Adult_female*, *Adult_male*, *Dead_mite*, *Immature*, and *Viable_egg* (see Figs 1 and 2). Detailed descriptions for each class are given in Fig 2. We did not capture images of the red to orange colored diapausing *T. urticae* because our interest here was in *in vitro* assays, which typically involve other life stages. Although the full dataset is publicly available, only a curated subset was used for training, validation, and testing. Images with uncertain annotations or nonstandard backgrounds (e.g., stickers) were excluded to ensure consistency and data quality. The Roboflow (Roboflow, Des Moines, Iowa) training and validation sets corresponding to model versions v209, v210, and v211 are available through RoboFlow [20]. Finalized and augmented test sets, which include images added after Roboflow versioning, are available through Zenodo here [21].

As is typical of mite populations, the dataset was strongly imbalanced toward the *Viable_egg* class (see Fig 1B,E). To mitigate this, we supplemented the dataset with synthetically generated images. Individual objects were segmented using the polygonal lasso tool in Adobe Photoshop (2024) and saved to class-specific folders. A custom JavaScript script (`generate_bounding_boxes.js`) was used to randomly place up to 50 augmented objects per image on randomly selected

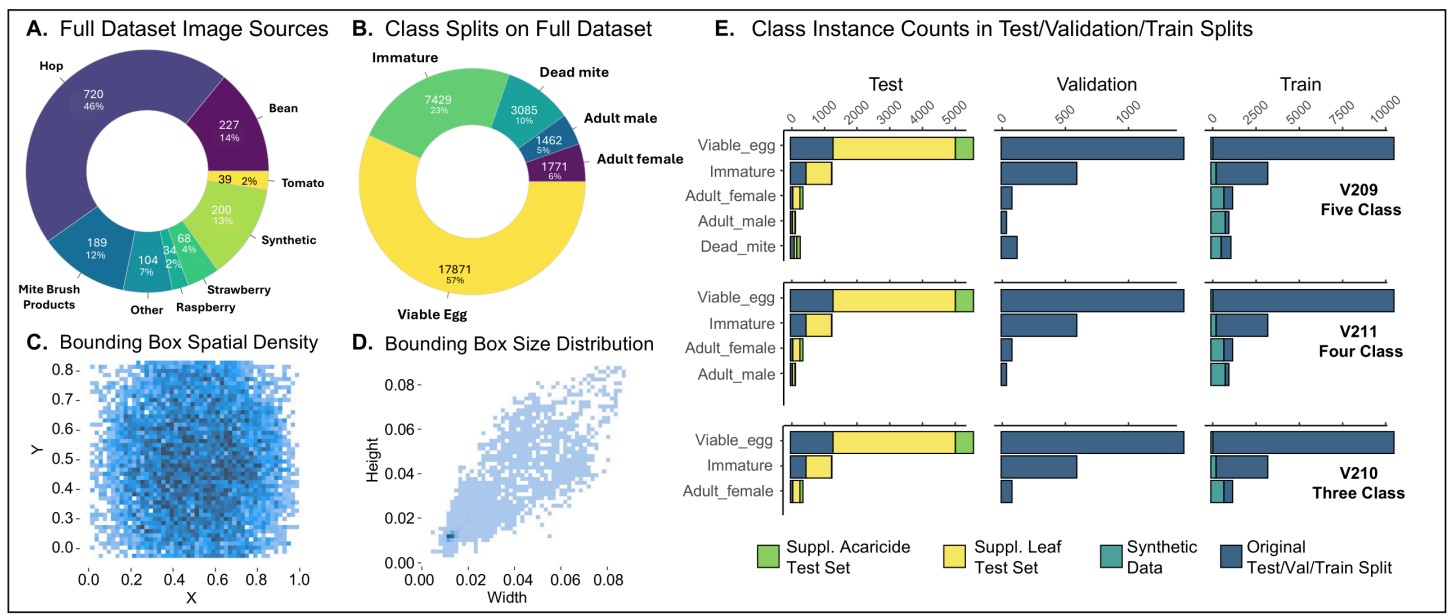

**Fig 1**. **Dataset splits and metadata. (A)** Distribution of diverse host sources for all of the images hosted on Roboflow. The "Other" category includes poppy, columbine, apple, pear, and nasturtium. **(B)** Distribution of class representation across the full dataset. **(C)** Heatmap showing the normalized center coordinates (x, y) of annotated bounding boxes in YOLO format. **(D)** Heatmap showing the normalized width and height of bounding boxes in YOLO format. Most annotations correspond to small objects (≤ 4% of the image in both dimensions).**(E)** Bar plots show the number of instances per class within each dataset split (columns: test, validation, training) and model version (rows: v209, v211, v210). Bars are stacked and color-coded by data source: original dataset (dark blue), synthetic data (teal), supplemental leaf test sets (yellow), and miticide trials (green).

backgrounds. Geometric augmentations included random rotation, scaling ($\pm$0.01), and position jitter, while color augmentations included saturation and brightness shifts ($\pm$10%). Output included a composite image and a corresponding XML file with bounding box annotations.

Image acquisition was performed using the Blackbird CNC Microscopy Imaging Robot (Version 1), developed through collaboration between Cornell University, the USDA-ARS Grape Genetics Research Unit (Geneva, NY), and Moblanc Robotics (Binéfar, Spain) [22,23]. The system automates acquisition of Z-stack images from up to 351 samples per tray using a Nikon Z 7II mirrorless camera paired with a Venus Optics Laowa 25mm ultra-macro lens set to 3.5× magnification. Z-range estimation is performed for each sample to ensure adequate coverage regardless of surface flatness. Z-stack images are acquired every 200 $\mu$m and merged using a custom Python script (`stackPhotosParallel.py`), which calls Helicon Focus (v8.1 from Heliconsoft, Kharkiv, Ukraine) for image stacking. Settings in Helicon Focus were as follows: method:B, depth map radius: 1, smoothing: 4, sharpness: 2. Each final image captured an area of 0.99 × 0.66 cm with a pixel resolution of 1.2 $\mu$m.

## Labeling

Bounding box labels were created using the bounding box tool in Roboflow Annotate. Labels were assessed independently by two experts to ensure label accuracy. In cases where there was disagreement between labelers on the biological sex or life stage, the image would be tagged as uncertain and was excluded from the sex differentiation dataset or removed all together.

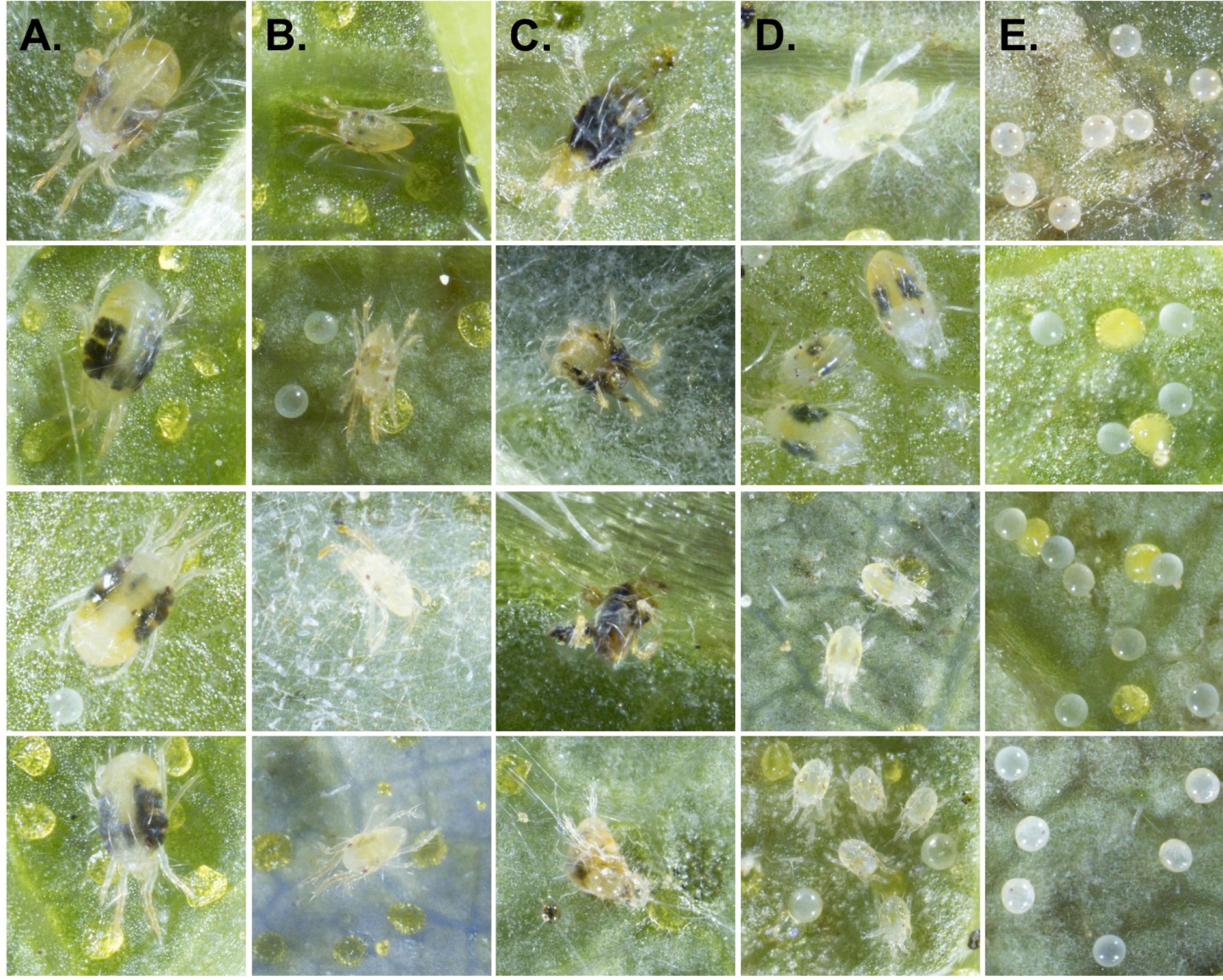

**Fig 2**. **Examples of each of the labeled classes. (A)** Mites measuring approximately 300–450 $\mu$m in length and 400–500 $\mu$m in width, with large, rounded abdomens, were annotated as *Adult_female*. **(B)** Those measuring approximately 200–300 $\mu$m in length and possessing tapered abdomens were annotated as *Adult_male*. **(C)** Mites were designated as *Dead_mite*s if they exhibited obvious signs of mortality, including ruptured bodies with internal contents exposed, desiccation, discoloration, or abnormal body posture (e.g., curled legs or unnatural orientation). **(D)** Individuals that did not fit any of the other categories—living deutonymphs, protonymphs, or larvae—were categorized as *Immature*. **(E)** An egg was classified as a *Viable_egg* if it was turgid, approximately 120-140 $\mu$m in diameter, clear to opaque or slightly orange in color, occasionally exhibiting two visible red eyespots. Note that the approximately circular, yellow objects in the second and third rows are glandular trichomes (lupulin glands) on a hop leaf.

## Model selection

YOLO (**Y**ou **O**nly **L**ook **O**nce) is a family of real-time object detection models known for their speed and accuracy [24]. Unlike two-stage detectors such as Faster R-CNN, which use a region proposal network before classification, YOLO performs object localization and classification in a single forward pass [25]. This one-stage architecture enables fast inference, making YOLO well-suited for real-time or high-throughput applications. Since its initial release in 2016 by Joseph

Redmon et al. [26], the YOLO architecture has undergone numerous revisions and improvements [27–34], with YOLO11 being released September 2024. YOLO11 introduces an enhanced backbone and neck architecture that improves feature extraction while reducing parameter count.

Like earlier versions, YOLO11 provides several model variants—ranging from compact (nano) to high-capacity (extra large)—all pre-trained on the Common Objects in Context (COCO) dataset, a widely used benchmark for object detection. In preliminary benchmarking experiments, larger YOLO variants consistently achieved higher detection accuracy than both smaller YOLO models, RT-DETR-L, Faster R-CNN (ResNet50 and MobileNetV3) under identical training conditions (see S1 Methods and S1 Table in S1 File). YOLO11-L offered the best balance of accuracy and inference speed and was therefore selected as the architecture for fine-tuning. This higher-capacity model enables improved generalization and fine-grained detection performance, which is particularly important for distinguishing small, visually similar arthropod life stages.

## Model training and evaluation

Model training was performed using NVIDIA A100 40GB or H100 80GB GPUs via Google Colab and the Oregon State University High Performance Computing Cluster. All models were trained on 1024×1024×3 images (padded to retain aspect ratio) to minimize information loss, given the small size of the target objects relative to the full image. Models were trained for up to 100 epochs or until validation loss plateaued for 25 consecutive epochs to prevent overfitting. Default YOLO11 hyperparameters and on-the-fly data augmentation settings were used for training (see the Ultralytics GitHub for implementation details). Hierarchical feature representations learned by YOLO11, illustrating the model's internal feature extraction and detection processes, are shown in S1 Fig in S1 File .

To assess model robustness and generalizability, we conducted $k$-fold cross-validation using the five-class configuration (v209). The dataset, excluding synthetic images from validation, was partitioned into five folds for cross-validation, with each image serving once as validation and four times for training. Mean performance metrics, including precision (Eq 1), recall (Eq 2), and mAP50 (mean average precision at an intersection over union threshold $\geq 0.50$; Eq 3), were calculated across all folds to summarize overall model performance. Intersection over union (IoU) is the area of overlap between a predicted bounding box and the corresponding ground truth box, divided by the area of their union (Eq 4). An mAP50 IoU threshold of 0.50 implies that a detection is considered a true positive if its predicted box overlaps the ground truth box by at least 50%. To assess the stability of model performance across folds, we computed the coefficient of variation for each performance metric, with a target threshold of less than 10%.

$$\text{Precision} = \frac{\text{True Positives}}{\text{True Positives} + \text{False Positives}} = \frac{|\text{Matched Predictions}|}{|\text{All Predictions}|} \tag{1}$$

$$\text{Recall} = \frac{\text{True Positives}}{\text{True Positives} + \text{False Negatives}} = \frac{|\text{Matched Predictions}|}{|\text{Ground Truth Objects}|} \tag{2}$$

$$\text{mAP} = \frac{1}{N} \sum_{i=1}^{N} \text{Average Precision}_i \tag{3}$$

$$\text{IoU} = \frac{\text{area of overlap}}{\text{area of union}} = \frac{|B_{predicted} \cap B_{groundtruth}|}{|B_{predicted} \cup B_{groundtruth}|} \tag{4}$$

After confirming the stability of cross-validation splits, the original dataset was partitioned into training (80%), validation (10%), and test (10%) sets. Three models were trained: one using all five-classes (v209), and two additional models with progressively fewer classes, created by sequentially removing the lowest-mAP50 performing classes (v210 and v211). To evaluate generalizability to novel conditions not encountered during training, the test set was supplemented post hoc with images from previously unseen host species, cultivars, and miticide treatment experiments. These supplemental images

were excluded from model training and hyperparameter tuning. To ensure the integrity of the evaluation, a custom script was used to confirm that no images from the training or validation sets were present in the test set.

Performance in object detection models like YOLO often degrades due to missed detections arising from high object density (crowding) and occlusion artifacts, including edge truncation, partial visibility, or full obstruction. We systematically evaluated how object density, spatial position, and diverse host background characteristics (e.g. occlusion from trichomes) affected detection performance across our three YOLO model versions: v209 (five-class), v210 (three-class), and v211 (four-class), using a shared test set with class labels adjusted accordingly. The five-class model (v209) included classes as noted previously. The streamlined four-class (v211) and three-class (v210) models omit the *Dead_mite* and/or *Adult_male* classes for applications where these categories are unnecessary (e.g., in fecundity assays).

Per-image precision and recall were calculated and stratified by class, host species, and cultivar (when applicable). Associations between object density and detection performance were analyzed using Pearson correlation and linear regression. To capture nonlinear effects, we fit generalized additive models (R package mgcv), including host or cultivar as fixed effects. Cultivars or species with fewer than five images were grouped under "other" to reduce the risk of overfitting. To evaluate model transferability across plant hosts of TSSM, we also assessed performance on test images from three previously unseen host species: apple, pear, and nasturtium. Each of the unseen species exhibit diverse foliar characteristics such as color, reflectance, presence of trichomes, and surface hair density (see Fig 3).

To complement the controlled test set evaluations and assess real-world utility, we next applied the trained models to biological data collected from *in vitro* assays adapted for use in the Blackbird platform. This allowed us to evaluate performance under practical assay conditions, where variation in leaf morphology or mite behavior may impact detection.

## Assessment of model performance in *in vitro* applications

**Mite source and rearing conditions.** Mites were originally obtained from two separate greenhouse populations ("OSU North" and "OSU West"), and males from the OSU North population were introduced every generation to prevent inbreeding depression. Colonies were maintained on surface-disinfested, young, fully expanded trifoliate-stage bean leaves (*Phaseolus vulgaris*, cv. 'Blue Lake') grown in a laboratory, placed in deep-dish Petri dishes (100 × 25 mm) atop sterile ddH$_2$O saturated paper towels. Paper towels were kept moist to maintain leaf turgor, and leaves were replaced every 5–7 days. Colonies were kept in a Percival growth chamber at 25,°C, 50–60% RH, and a 14:10 h light:dark cycle,

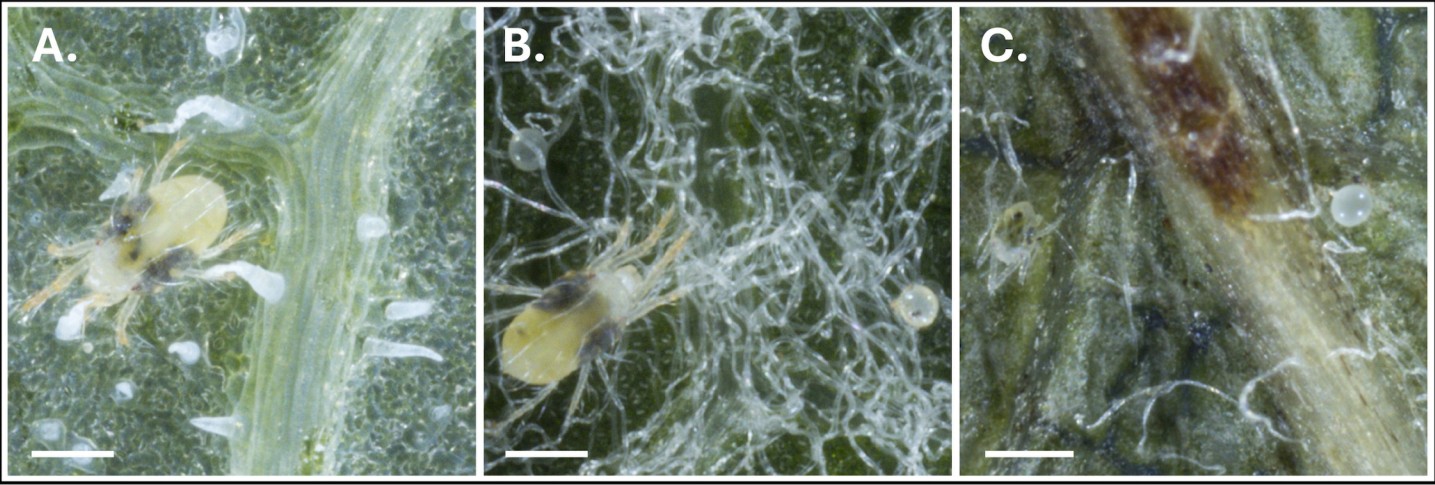

**Fig 3**. **Examples of differing host topographical features in our unseen host test set. (A)** nasturtium, **(B)** apple, **(C)** pear. Scale bars are 300 $\mu$m.

maintained at moderate densities (approximately 1–2 adult females per $cm^2$), and checked regularly for contamination. Prior to the laboratory assays, the mite colonies had undergone at least three generations under these conditions.

**Model performance during *in vitro* miticidal assays.** We applied our five-class model (v209) to data generated from an *in vitro* miticidal assay to assess model performance in assessing viability and fecundity post-pesticide exposure. Bean leaf disks (0.8 cm diameter) were immersed in bifenazate solutions formulated as FLORAMITE SC (OHP, Bluffton, SC) at concentrations of 0 (control), 3.125, 6.25, 12.5, 25, 50, 100, and 200 ppm, air-dried, and placed adaxial-side down onto 0.7% water-agar trays. Three to five motile mites were transferred onto each disk, with seven replicate disks per treatment. To prevent mite escape, agar moats were created by pouring 0.7% agar into Blackbird tray molds, allowing it to solidify, and flooding the moats with sterile water (see Fig 4). Mite viability, calculated as the proportion of dead individuals relative to the total number of live and dead mites, and fecundity, measured as the number of eggs laid per gravid female, were assessed both manually and using the object detection model five days after treatment. Model inferences were compared to ground-truth annotations to evaluate stage-specific classification accuracy and biological relevance. To quantify bifenazate effects and assess model accuracy, we fit binomial generalized linear models (GLMs) to mortality data, which represent binary outcomes (dead versus alive), and negative binomial GLMs to fecundity data. Dose–response curves were used to estimate $LC_{50}$ values, and precision–recall metrics were computed to evaluate detection performance across mite classes. All statistical analyses and visualizations were run in R (v4.3.0) using standard packages including `ggplot2`, `lme4`, `drc`, and `emmeans` [35–38].

**Model performance during assessment of mite fecundity on hop cultivars.** To further evaluate the utility of our detection system, we conducted a second case study focused on assessing mite fecundity across different hop cultivars. Specifically, we tested whether our three-class model (v210) could detect differences in oviposition rates among 'Cascade', 'Nugget', and 'Pacific Gem'. Gravid females used in this experiment were obtained by allowing gravid females to oviposit for 24 h, after which the eggs were reared on bean leaves with mature males until adulthood (approximately

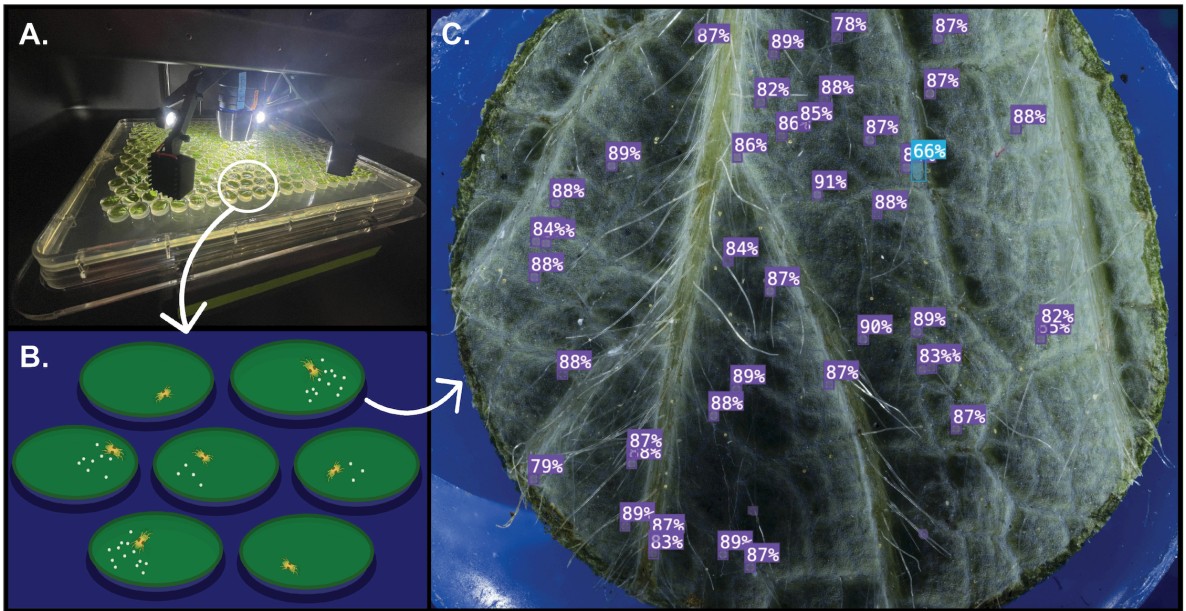

**Fig 4**. **Overview of TSSM assay integration into the Blackbird platform and subsequent computer vision detection. (A)** The Blackbird platform imaging mite-infested leaf disks. **(B)** Schematic of experimental setup: individual leaf disks are placed on 0.7% water agar and surrounded by sterile water moats to minimize mite escape. **(C)** Example output from model inference on a strawberry leaf showing automated object detection. Purple labels indicate predicted *Viable_egg* instances and the blue label indicates an *Adult_male*. Percentages represent model confidence scores.

12 to 14 days later under the same growth chamber conditions described above). Newly emerged females were transferred to leaf disks of each hop cultivar 24 h post eclosion and allowed to oviposit over a 5-day period. An equal number of control disks, maintained under identical conditions but without gravid females, were included to quantify false positive detections. Trays were incubated for five days in the same conditions as above. For each cultivar, five leaf disks were tested, and the experiment was repeated across three independent runs. Fecundity was quantified as the number of eggs laid per day and summarized for each leaf-disk replicate by calculating the area under the curve (AUC) across the five days. Prior to model fitting, AUC values were tested for normality within each cultivar using the Shapiro-Wilk test ($p>0.05$), confirming that the normality assumption was met. We then fit a linear mixed model with *cultivar* as a fixed effect, *run* as a random effect, and AUC as the response variable. Post hoc pairwise comparisons were conducted using Tukey's HSD.

## Results

### Model performance

The five-fold cross-validation showed consistent performance across folds, with mean precision of 0.810, recall of 0.749, mAP50 of 0.807, and mAP50-95 of 0.499 (see S2 Table in S1 File). Class-wise performance metrics are reported in S3 Table in S1 File. Precision and recall were highest for the *Viable_egg* and *Adult_female* classes, while performance was lowest for the *Dead_mite* class likely due to variability in postmortem morphology and under-representation in training.

The five-class model (model v209) showed the lowest performance, primarily due to poor recall and high confusion in the underrepresented *Dead_mite* class (see Fig 5A, Table 1). Although *Viable_egg* and *Adult_female* detections remained strong (recall: 0.92 and 0.87), *Dead_mite*s were frequently missed (21% false negatives) or misclassified as *Adult_female*s (10%), reflecting morphological ambiguity and limited training examples.

The four-class model (model v211) maintained strong recall on *Viable_egg*s (0.92) and *Adult_female*s (0.90), but had reduced recall for *Immatures* (0.87) and *Adult males* (0.80). Confusion between these two visually similar classes increased, likely due to labeling ambiguity in late-stage immatures. *Immatures* also showed elevated background confusion (0.37, n=46 false positives) (see Fig 5B).

The three-class (model v210) achieved the highest recall for *Viable_egg* (0.93), *Adult_female* (0.92) and *Immature* (0.89). This model exhibited minimal background confusion; however, false positives were still notable for *Viable_egg*

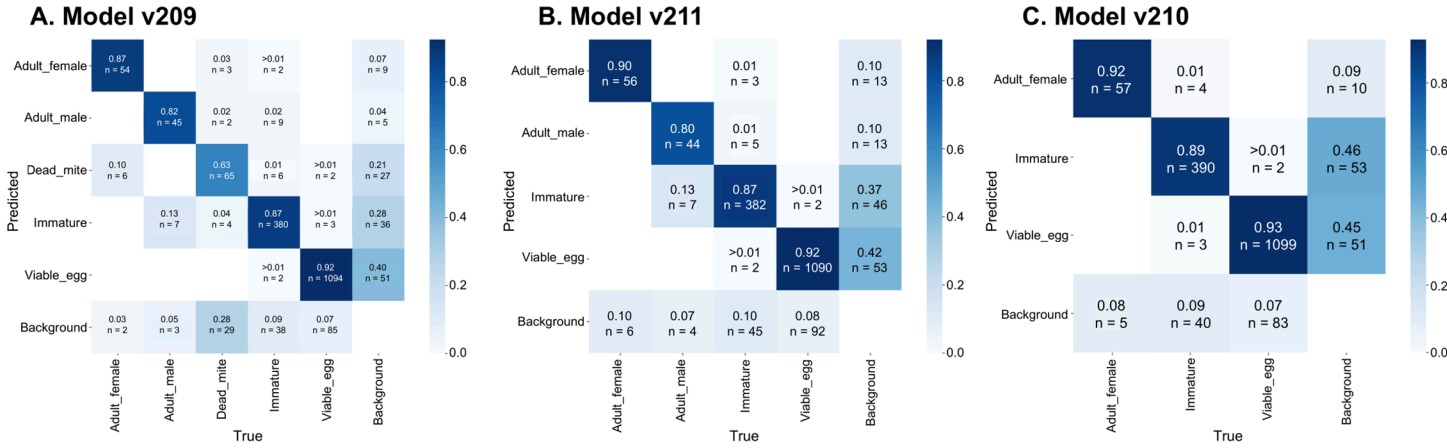

**Fig 5**. **Confusion matrices for the three object detection models. (A)** v209 (five-class), **(B)** v211 (four-class), and **(C)** v210 (three-class). Each matrix represents normalized classification performance for predicted versus true object classes. The value in each cell reflects the proportion of true instances (rows) predicted as a given category (columns), with raw object counts labeled below each proportion. These matrices highlight common misclassifications, such as confusion between *Viable_egg* and *Immature*, particularly in the more complex class sets. Models were evaluated across a diverse image set spanning multiple host substrates.

**Table 1.** Test set performance of the three evaluated models.

| Model | Classes | Precision | Recall | mAP50 | mAP50–95 | Instances |
|---|---|---|---|---|---|---|
| 209 | *Adult_female, Adult_male, Dead_mite, Immature, Viable_egg* | 0.824 | 0.743 | 0.788 | 0.532 | 2093 |
| 210 | *Adult_female, Immature, Viable_egg* | 0.875 | 0.871 | 0.883 | 0.601 | 1938 |
| 211 | *Adult_female, Adult_male, Immature, Viable_egg* | 0.903 | 0.835 | 0.871 | 0.576 | 1949 |

(0.45, n=51) and *Immature* (0.46, n=53), largely due to misidentification of leaf structures (e.g., developing lupulin glands on hop leaves) and adult males in the images being classified as *Immatures* (see Fig 5C).

Overall, performance in the three models improved as the number of classes decreased. While the three-class model (v210) yielded the clearest class separation, the five-class model (v209) struggled with performance on *Dead_mite*, an ambiguous and underrepresented class. These findings highlight the trade-off between class specificity and detection reliability in multiclass mite detection. Test images showcasing each model's performance can be observed in Fig 6 and S2-S3 Figs in S1 File and model weights are available through Zenodo [39].

Considering all host species in the leaf test set, per-image recall varied significantly by host (see Fig 7). While most of this variation in recall likely reflects differences in training data representation, e.g., recall was significantly lower on tomato (*effect* = –0.128, $p = 1.23 \times 10^{-7}$) and raspberry (*effect* = –0.137, $p = 4.84 \times 10^{-7}$), which had relatively few training images, other patterns suggest additional influencing factors. Notably, recall was slightly but not significantly lower on hop compared to bean (*effect* = –0.032, $p = 0.075$), despite hop being heavily represented in the training set (482 images). This suggests that factors beyond training representation may have a subtle influence on performance. Recall had a non-linear decline at high object densities ($p = 0.030$; see S4 Fig in S1 File), remaining stable at low-to-moderate counts but decreasing notably above 80 objects per image. This drop in recall likely reflects increasing occlusion or crowding effects above about 80 objects per image, which can obscure object boundaries and reduce detection sensitivity.

In contrast to recall, per-image precision showed minimal variation across host species and was not significantly influenced by object density (see Fig 7). The only significant host effect was a slight decrease in precision on tomato compared to bean (–0.025, $p = 0.022$); differences for hop, raspberry, and strawberry were not significant. Additionally, the

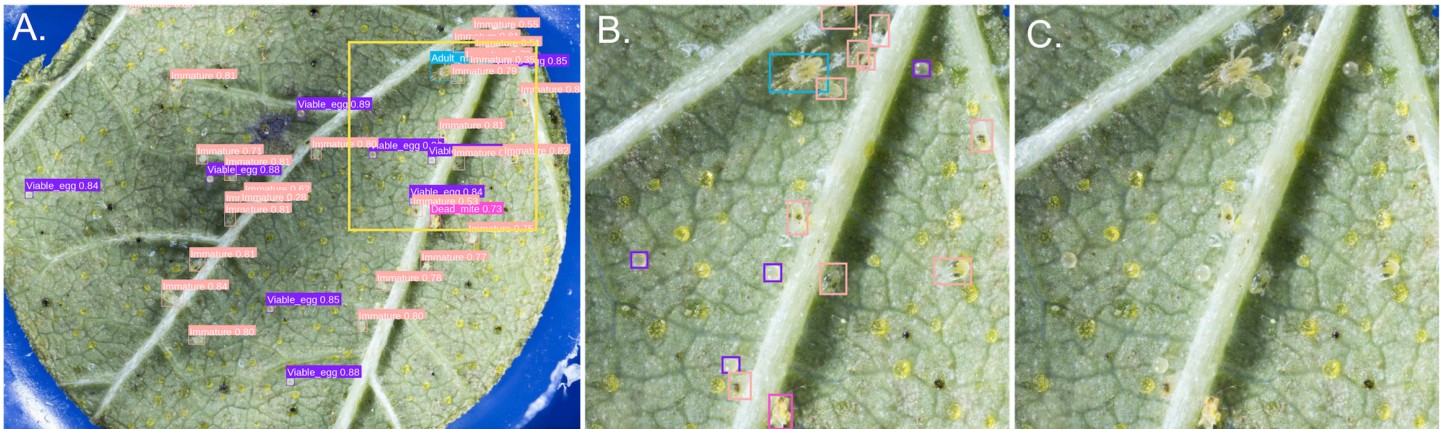

**Fig 6.** **Example performance of the five-class detection model (v209) on a test-set image.** **(A)** Model inference results (IoU = 0.5, confidence threshold = 0.5) showing detected objects with labels and confidence scores. Detected classes are color-coded as follows: *Adult_male* (teal), *Immature* (salmon), *Viable_egg* (purple), and *Dead_mite* (hot pink). **(B)** Magnified view of the region outlined in panel A, illustrating the model's ability to detect partially occluded individuals and to distinguish immatures and eggs from morphologically similar lupulin glands. **(C)** The same magnified region shown without labels for visual comparison.

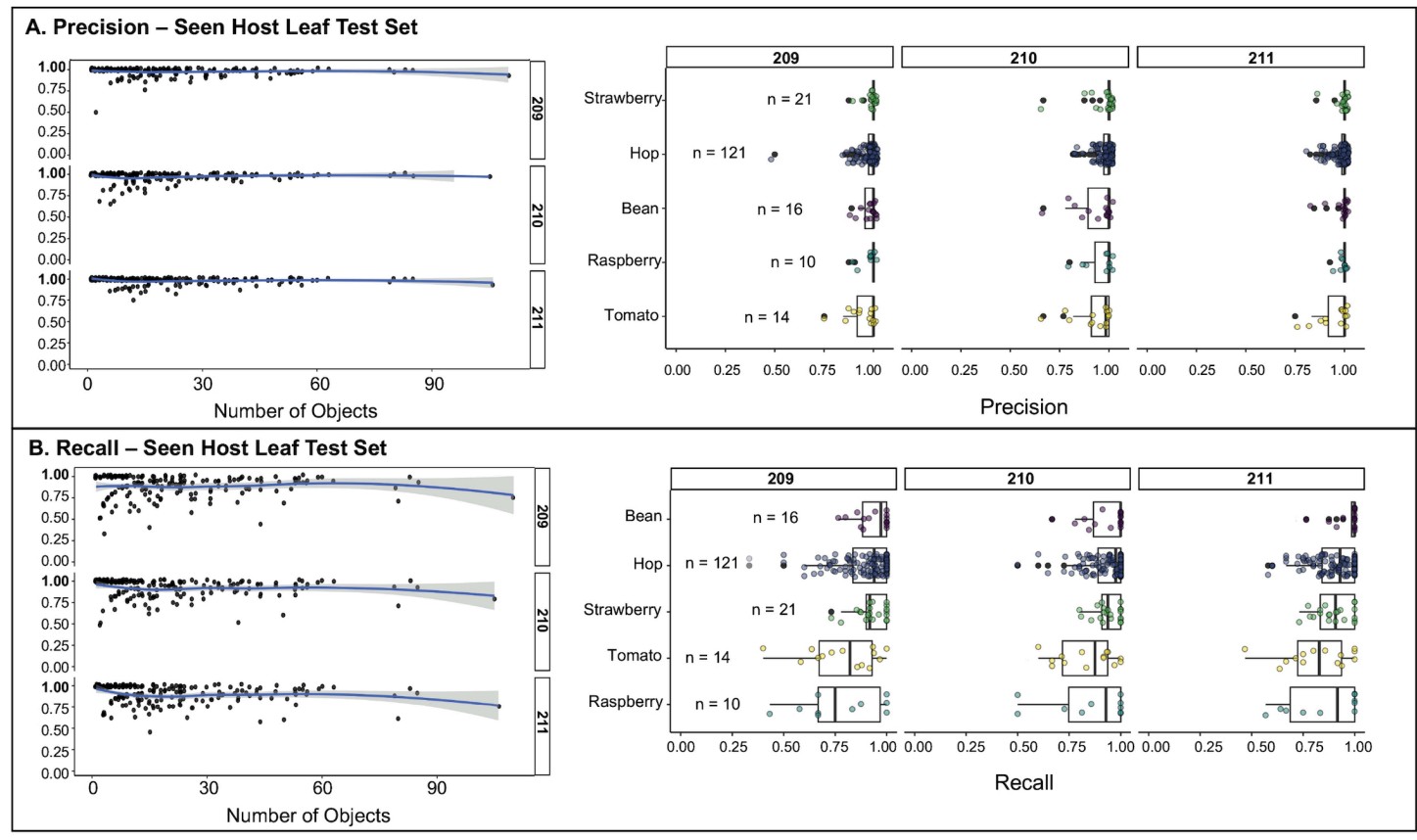

**Fig 7**. **Model performance against expansive supplemental leaf test set (190 images; excluding unseen hosts)**. **(A,B Left)** Each point represents a single image from the seen-host leaf test set, showing the relationship between total object count and **(A)** precision or **(B)** recall. LOESS smoothing curves are overlaid to highlight performance trends. All models exhibited consistently high precision and recall across a wide range of object densities, demonstrating robust detection performance in dense and sparse conditions. **(A,B Right)** Performance metrics are shown for each model across five previously seen host species (Bean, Hop, Raspberry, Strawberry, and Tomato). Each point represents a single image, with box plots summarizing the distribution for each group. Host box plots are displayed in ascending order, sorted by the grouped model mean recall or mean precision.

smooth term for object count was not significant ($p = 0.589$), suggesting that increasing the number of objects in an image did not meaningfully impact precision. These results suggest that while object crowding and host features may reduce the likelihood of detecting all true objects (recall) but appear to have relatively little effect on false positives (precision).

When evaluating model performance on unseen hosts not used in model training, Wilcoxon rank-sum tests revealed significantly higher median recall on pretrained hosts ($p = 4.32 \times 10^{-5}$), while the difference in precision was not significant ($p = 0.381$). These results suggest that recall is more sensitive when TSSM are present on host types not included during model training, whereas precision remains relatively robust across host types (see S5 and S6 Figs in S1 File).

### Assessment of model performance in *in vitro* applications

**Model performance during *in vitro* miticidal assays.** Both manual and model-derived annotations from the five-class model (v209) detected significant, dose-dependent effects of bifenazate on mite viability and fecundity (55 images, 709 class instances). Binomial GLMs confirmed that mite mortality increased with log-transformed pesticide concentration for both ground-truth ($\beta = 0.638$, SE = 0.115, $p = 2.96 \times 10^{-8}$) and model-derived annotations ($\beta = 0.479$, SE = 0.114,

$p = 2.49 \times 10^{-5}$). However, $LC_{50}$ estimates diverged substantially: 26.3 ppm for manual counts (SE = 7.57; 95% CI: 11.1–41.5; $p = 0.0010$) versus 212.4 ppm for model-derived counts (SE = 138.6; 95% CI: –65.5 to 490.3; $p = 0.16$) (see Fig 8). This divergence was primarily due to underestimated mortality caused by misclassification of dead mites as motile stages (39/97) and missed detections (8/97). Overestimation of survivorship could limit the model's accuracy in estimating pesticide efficacy. Despite this discrepancy, the model demonstrated strong overall detection performance across assay images, with a mean precision of 0.845 and recall of 0.868 when aggregating across all mite classes.

Fecundity estimates post-miticide treatment were more consistent across methods. Negative binomial models (with offsets for adult female counts) revealed significant declines in egg production with increasing bifenazate concentration: 38.9% per log-unit for model-derived data ($\beta = -0.389$, SE = 0.059, $p = 5.86 \times 10^{-11}$) and 31.7% for manual counts ($\beta = -0.317$, SE = 0.059, $p = 7.78 \times 10^{-8}$). These results indicate that model v209 reliably captures biologically meaningful reductions in fecundity, despite underestimating acute mortality.

**Model performance during assessment of mite fecundity on hop cultivars.**  Using a subset of the leaf fecundity images (105 images, 2,825 class instances), we evaluated the three-class model (v210) and found it achieved high recall (0.915–0.972) and mAP50 (0.952–0.977) across the three hop cultivars, supporting its suitability for object detection in hop fecundity assays. Strong agreement between predicted and ground truth counts across cultivars, model versions, and mite classes further supports the model's performance in this application (see S7 Fig in S1 File); linear models fit to each combination of model and cultivar yielded coefficients of determination ($R^2$) $\geq 0.98$, with the lowest values observed in

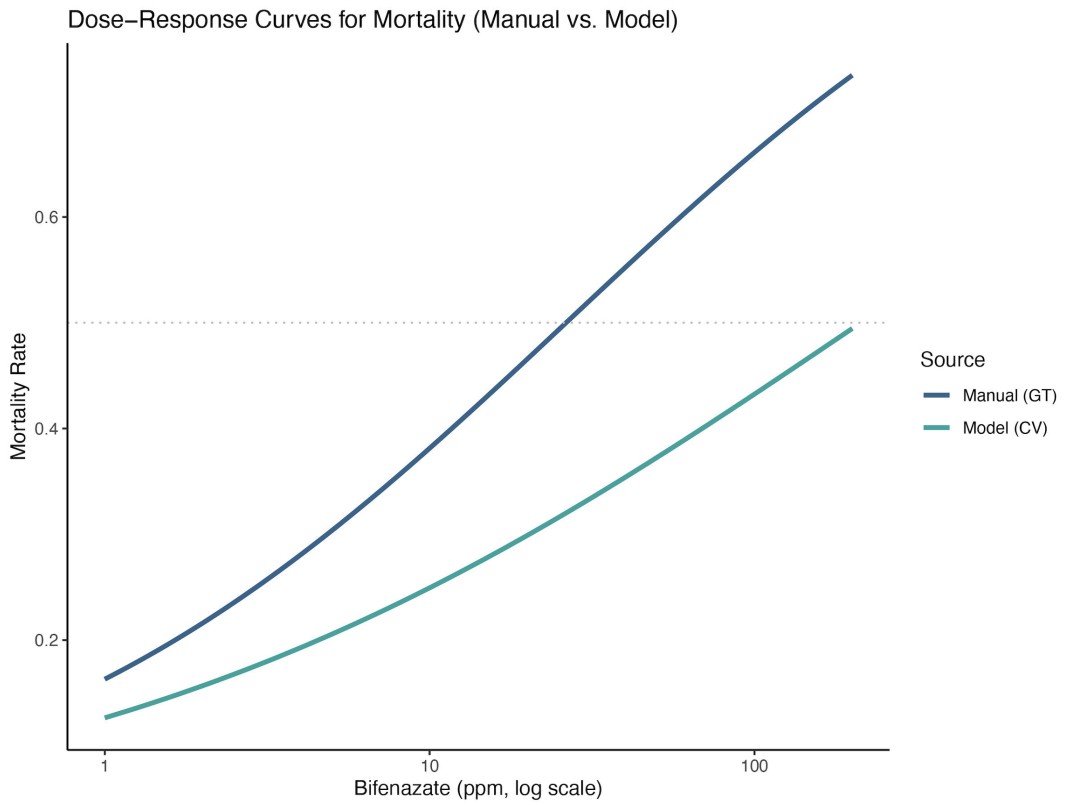

**Fig 8**. **Dose–response curves for mite mortality to bifenazate, comparing manual scores and v209 model predictions.** Mortality rate is shown as a function of bifenazate concentration (ppm, log scale), with the dotted horizontal line indicating $LC_{50}$. Substantial discrepancies between v209 model inferences and ground truth (GT) ratings suggest that still-image RGB object detection models may have limited suitability for reliably detecting recent mite mortality.

Pacific Gem. Thus, we ran the entire dataset (250 images, 6,283 class instances) through inference with model v210 and detected significant cultivar effects on TSSM fecundity: Cascade exhibited a significantly higher AUC than Nugget (Tukey HSD, $p = 0.0066$), while differences between Cascade and Pacific Gem ($p = 0.0717$) and between Nugget and Pacific Gem ($p = 0.6219$) were not significant (see Fig 9).

## Discussion

We developed an extensive and publicly available image dataset of annotated images of TSSM life-stages from diverse host plants and cultivars. We leveraged this image dataset to develop an automated microscopy and computer vision pipeline to improve the throughput and reproducibility of *in vitro* assays for the two-spotted spider mite. Our approach addresses key limitations of manual evaluation methods, particularly labor requirements and rater subjectivity, and achieves human-level accuracy in detecting and quantifying TSSM life stages when applied to data similar to the training set.

To understand the limitations in our system, we examined the correlation between object density and model performance which provided practical considerations for future assay design, specifically regarding mite density thresholds beyond which performance deteriorates. As expected, we found that recall for our models drops when there are more

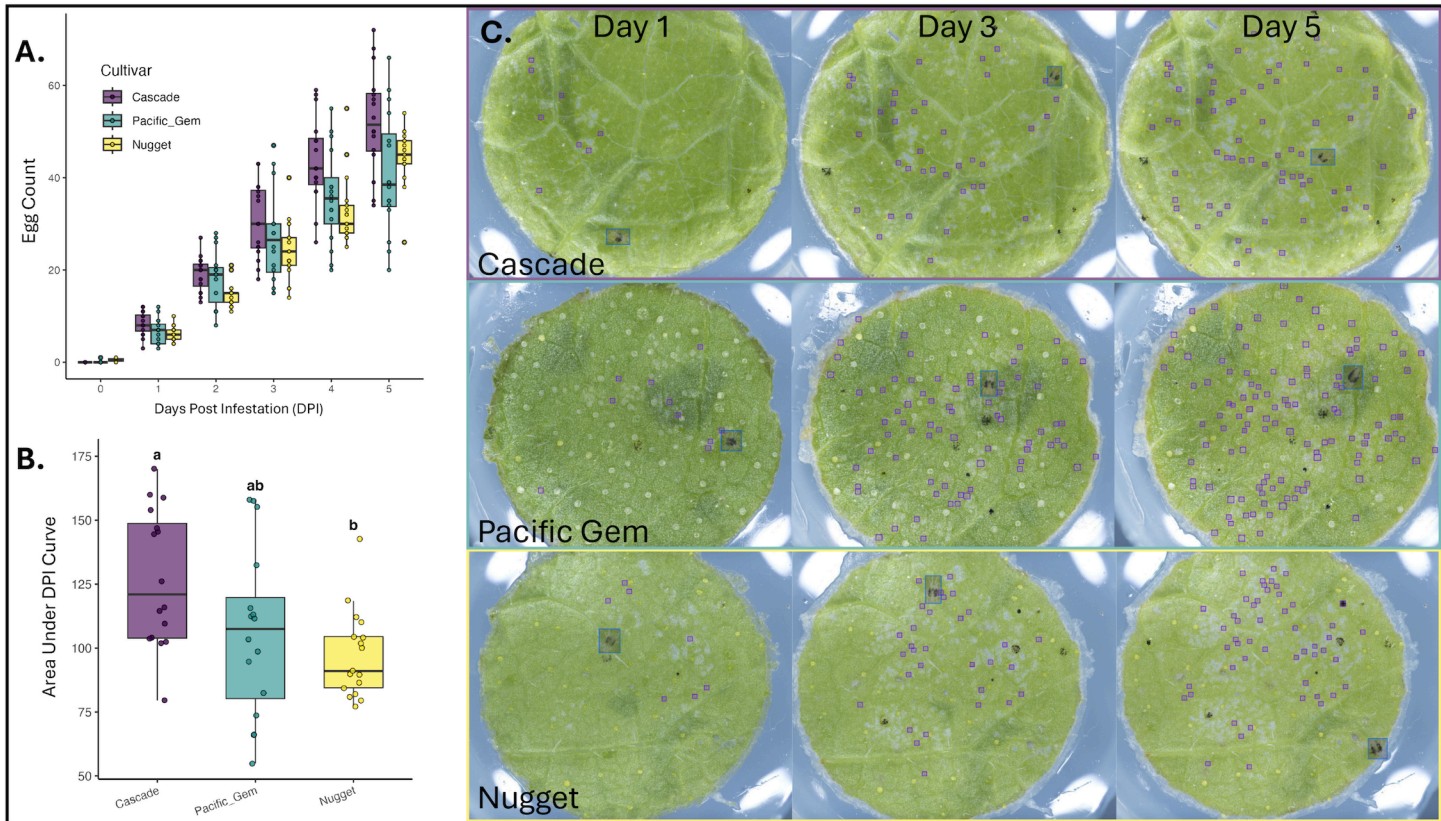

**Fig 9**. **Comparison of mite fecundity across hop cultivars.** Computer vision–based egg counts were used to quantify fecundity over a five-day period on three hop cultivars: Cascade, Nugget, and Pacific Gem. **(A)** Daily egg counts showed cultivar-specific differences in oviposition. **(B)** Area under the curve (AUC) of daily counts was calculated for each replicate to summarize fecundity. A linear model with cultivar and experimental run as fixed effects revealed significant differences in fecundity between cultivars (Tukey's HSD, $p<0.05$). Error bars indicate standard error of the mean. **(C)** Representative images that illustrate the variability in fecundity and leaf features between the cultivars.

than approximately 80 objects in an image due to partial or full object occlusion. Recall also dropped when models were applied to host plants not used in model training. However, precision was robust to both the number of objects in an image and host background. Therefore, care is needed when applying our models for quantitative assessment of mite life stages when motiles or eggs are extremely dense or when other objects such as environmental debris may occlude mites.

Potential limitations of our system are also challenges encountered by human raters during microscopic assessment of mites (see Fig 10). For example, distinguishing late-stage deutonymphs from adult females is difficult due to their similar size and coloration. Model accuracy declined in cases of partial or full occlusion, motion blur caused by mite movement during image capture, or when background structures—such as trichomes or developing glands—closely resembled target classes. Some of these limitations can be averted by sample handling procedures, careful consideration of experimental design, and additional model training. For example, we observed that pre-chilling trays at 4°C for one hour before

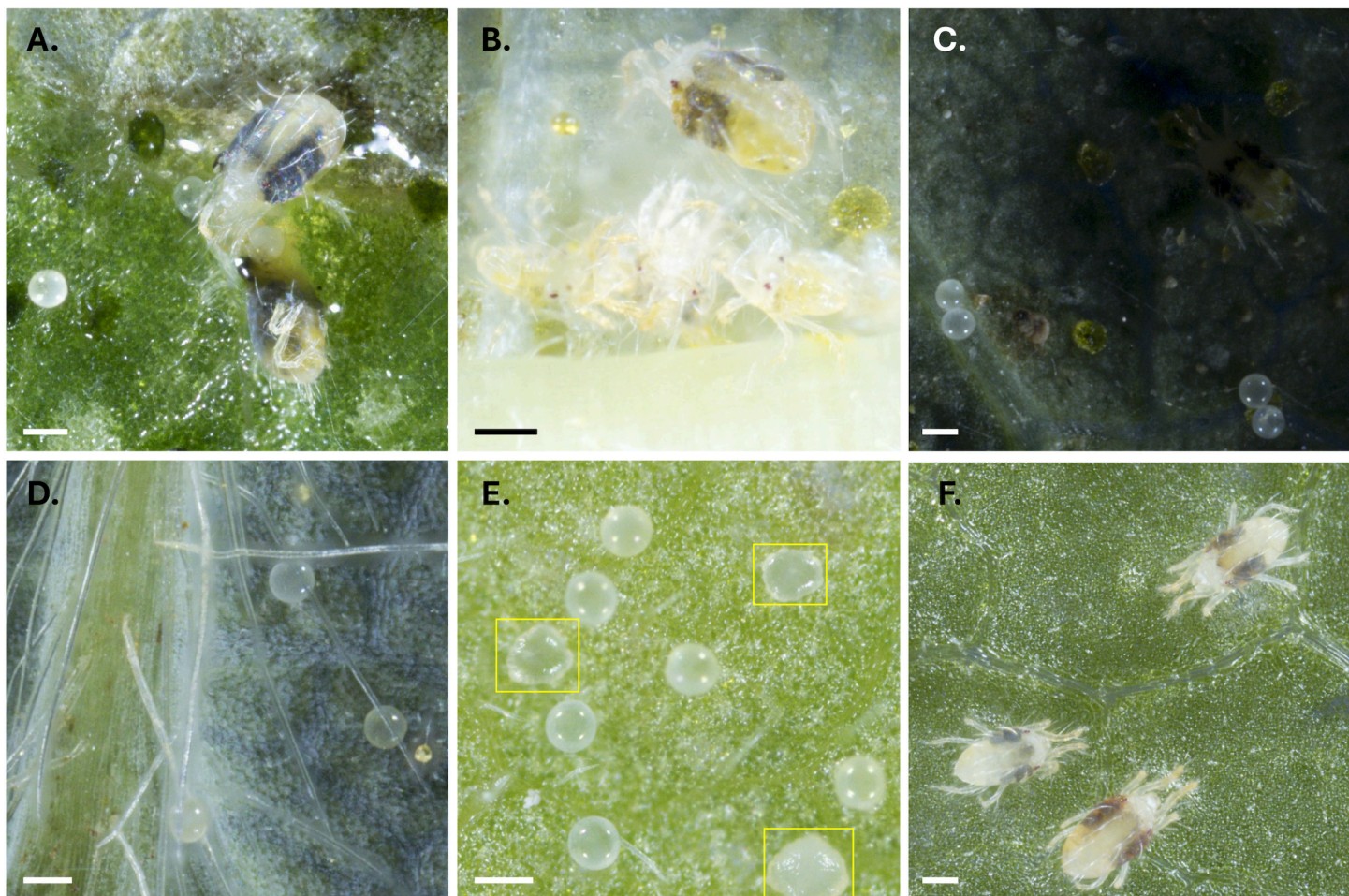

**Fig 10**. **Examples of challenging scenarios for model detection.** Representative cases where a model underperformed due to biological and technical complexities. **(A)** Crowding and z-axis occlusion among object classes. **(B)** Motion blur and crowding during active mite movement (more common with males). **(C)** Shadowing artifacts caused by leaf cupping. **(D)** Leaf features such as veins and trichomes occluding objects. **(E)** Confounding leaf structures, such as developing lupulin glands on hop leaves (in yellow boxes), visually resembling mite eggs. **(F)** Mites treated with miticide imaged 5 days post-application still appear viable, showing minimal discoloration and leg curling, confounding viability classification. Scale bars are 150 $\mu$m.

imaging substantially reduced mite movement and minimized motion-related artifacts. We also observed that model performance on new host plant species improved even with limited additional training data, suggesting adaptability with modest fine-tuning. While our models performed robustly across a wide range of conditions, human oversight remains important for inference in visually complex, ambiguous, or unfamiliar contexts (e.g., novel hosts, shaded objects, or partially visible mites). In such edge cases, human assessors can draw on contextual cues and biological experience to make judgments that models may miss.

We demonstrated that the Blackbird imaging system and our models were suitable for life stage identification and assessment of TSSM fecundity, and also showed that they may not be suitable for assessment of mortality. The relatively lower performance of the *Dead_mite* class in both the test set and miticide assay underscored the difficulty of mite mortality assessment with computer-vision models. Mortality assessment of *T. urticae* in bioassays typically involves physically touching each mite with a fine paint brush or other object under magnification. Individuals with no response are presumed to be dead [40]. As demonstrated in Fig 10F, mites recently treated with miticide could be visually indistinguishable from live individuals, especially shortly after death. This limitation is inherent to all RGB still image-based mite identification and enumeration platforms. Future improvements could incorporate object tracking or temporal analysis to better detect subtle movements to more accurately assess mite viability [41,42]. While our Roboflow deployed models support real-time detection and tracking of mite movement, the Blackbird software is not currently configured for video streaming or capture. We note that in many bioassays, mortality is not an important metric; consequently, we developed the four-class model for applications where enumeration of dead mites is not essential. Utilization of the four-class model (v211) improved overall performance as compared to the five-class model v209 (five-class test mAP50: 0.788; four-class test mAP50: 0.871).

A salient application of our automated assay pipeline is in breeding for host resistance to TSSM based on antibiosis. Such breeding programs typically require large populations and multi-generational selection, which are often constrained by the high cost of skilled labor for manual bioassays. Antibiosis can manifest through reduced fecundity, prolonged egg and nymphal development, or lower population growth rates [43–46]. The Blackbird imaging system, integrated with our computer vision pipeline, quantifies these traits accurately with minimal human effort, as demonstrated in our proof-of-principle experiment comparing mite fecundity across three hop cultivars. Once mites are transferred, hands-on time is limited to placing trays in the robot and initiating imaging. Each image stack required approximately 15 s for capture and ~10 s for focus stacking, yielding a throughput of roughly 140 disks h$^{-1}$. Automated detection and classification were rapid: inference required 2–5 s per image on a 2021 MacBook Pro CPU and <0.05 s on an NVIDIA A100 GPU. Compared with manual counting—which typically takes 30 s to 5 min per leaf disk and is susceptible to inter-rater variability and fatigue—the automated pipeline achieves at least an order-of-magnitude improvement in total throughput and a 1,000–10,000-fold reduction in per-sample analysis time, while substantially reducing hands-on labor and allowing researchers to focus on higher-level experimental tasks. As noted previously, achieving optimal performance on other host plants may require additional model training with host-specific data, but the present framework substantially reduces both the time and expertise required for large-scale quantitative *T. urticae* assays.

Efforts were also undertaken to implement the imaging pipeline for material collected from field-derived hop leaves using a mite brushing machine and funnel to concentrate the brushing by-products (see S8 Fig in S1 File). However, adapting field collection methods to this system introduced several challenges, including occlusion from other arthropods, occlusion from debris, mite eggs adhering to the funnel sidewalls, and mite fatalities caused by brushing trauma. These issues compromised the accuracy and reliability of our computer vision model, which is best suited for *in vitro* applications with *T. urticae*. We were able to overcome some of these challenges by imaging entire leaves in sections (similar to scanning), but this approach requires acquisition of ≥ 100 images per sample and thus is more time-consuming than manual rating (see S9 Fig in S1 File).

While not tested here, the pipeline could be adapted for ecological assays involving predation or parasitism. Desiccated or partially consumed mites would likely be classified as *Dead_mite*, whereas recently attacked but still intact individuals

would likely be classified according to their life stage. Incorporating such cases into model training could enable future extensions for quantifying predation rates or trophic interactions within integrated pest management studies.

The Blackbird platform could also be adapted to support host-preference (antixenosis) assays, in which TSSM are given a choice between cultivars positioned around infested leaf disks [11]. Integrating time-lapse imaging, image registration, and mite tracking could substantially enhance the assay's utility and represents a promising direction for future research.

## Conclusion

In this study, we developed and validated a high-throughput image analysis pipeline for TSSM *in vitro* phenotyping, integrating automated microscopy with computer vision object detection. Our models achieved human-level accuracy in identifying mite life stages under controlled conditions and proved effective in practical applications such as fecundity assays across host cultivars. By reducing labor demands and increasing assay reproducibility, this system offers a scalable alternative to manual scoring, with particular relevance for resistance screening in breeding programs where fecundity or the total number of motile mites predicts host resistance.

## Supporting information

**S1 File. Supplemental methods, figures, and tables.** This file contains all supporting methods, figures (S1–S9 Figs), and tables (Tables S1–S5). S1 Fig presents hierarchical feature representations and detection outputs from the YOLO11-L model; S2 Fig and S3 Fig illustrate example performance of the four-class and three-class detection models, respectively; S4 Fig shows generalized additive model smooth terms for recall and precision; S5 Fig summarizes precision and recall on previously unseen hosts; S6 Fig compares model performance on pretrained versus naïve hosts; S7 Fig shows predicted versus ground-truth mite detections for the fecundity study subset; S8 Fig documents field-collected mite imaging adaptations; and S9 Fig displays a stitched leaf-scan example using the Blackbird imaging system. Tables S1–S5 provide detailed benchmarking results across model variants, cross-validation folds, per-class metrics, and resolution limit testing.
(DOCX)

## Acknowledgments

The authors extend gratitude to Navneet Kaur and Alison Willette for providing mite specimens for imaging. We also thank the Cornell AgriTech, USDA-ARS GGRU, and Moblanc Robotics teams: Yu Jiang, Lance Cadle-Davidson, Anna Underhill, Javier Moreno, and Dani Martinez for providing our laboratory with the Blackbird imaging robot and assisting us along the way.

## Author contributions

**Conceptualization:** Michele S. Wiseman, Joanna L. Woods, Briana J. Richardson, David H. Gent.

**Data curation:** Michele S. Wiseman, Joanna L. Woods, Carly R. Hartgrave, Briana J. Richardson.

**Formal analysis:** Michele S. Wiseman.

**Funding acquisition:** David H. Gent.

**Investigation:** Michele S. Wiseman.

**Methodology:** Michele S. Wiseman.

**Project administration:** David H. Gent.

Resources: David H. Gent.

Software: Michele S. Wiseman.

Supervision: David H. Gent.

Visualization: Michele S. Wiseman.

Writing – original draft: Michele S. Wiseman.

Writing – review & editing: Michele S. Wiseman, Joanna L. Woods, Carly R. Hartgrave, Briana J. Richardson, David H. Gent.

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
