## [Decision Letter · Decision Letter 0]

6 Oct 2025

PONE-D-25-49646Automated detection and quantification of two-spotted spider mite life stages using computer vision for high-throughput in vitro assaysPLOS ONE

Dear Dr. Gent,

Thank you for submitting your manuscript to PLOS ONE. After careful consideration, we feel that it has merit but does not fully meet PLOS ONE’s publication criteria as it currently stands. Therefore, we invite you to submit a revised version of the manuscript that addresses the points raised during the review process.

We look forward to receiving your revised manuscript.

Kind regards,

Muhammad Asif Qayyoum, PhD

Academic Editor

PLOS ONE

Journal Requirements:

The authors received a Blackbird Imaging Robot through a sub-award from the USDA ARS AI Innovation Fund Cooperative Agreement (No. 0440860).

This research was conducted in support of United States Department of Agriculture CRIS project 2072-21000-061-000-D.

4. Please expand the acronym “USDA ARS” (as indicated in your financial disclosure) so that it states the name of your funders in full.

The authors extend gratitude to Navneet Kaur and Alison Willette for providing mite specimens for imaging. We also thank the Cornell AgriTech, USDA-ARS GGRU, and Moblanc Robotics teams: Yu Jiang, Lance Cadle-Davidson, Anna Underhill, Javier Moreno, and Dani Martinez for providing our laboratory with the Blackbird imaging robot and assisting us along the way. This research was conducted in support of United States Department of Agriculture CRIS project 2072-21000-061-000-D.

The authors received a Blackbird Imaging Robot through a sub-award from the USDA ARS AI Innovation Fund Cooperative Agreement (No. 0440860).

This research was conducted in support of United States Department of Agriculture CRIS project 2072-21000-061-000-D.

6. We notice that your supplementary tables are included in the manuscript file. Please remove them and upload them with the file type 'Supporting Information'. Please ensure that each Supporting Information file has a legend listed in the manuscript after the references list.

Additional Editor Comments:

Subject: Decision on Manuscript ID PONE-D-25-49646 - Minor Revision

Dear Authors,

Manuscript Title: "Automated detection and quantification of two-spotted spider mite life stages using computer vision for high-throughput in vitro assays"

Thank you for submitting your manuscript to PLOS ONE. It has now been assessed by three expert reviewers, and their comments are provided below.

I am pleased to inform you that the reviewers find your work on applying computer vision to spider mite life stage detection both valuable and timely. They acknowledge the potential of your system and the overall soundness of your approach.

Based on the reviewers' feedback and my own evaluation, I have decided that your manuscript requires Minor Revision before it can be accepted for publication. The reviewers have raised several constructive points that, once addressed, will significantly strengthen the impact and clarity of your study.

The key points to address are:

Broader Context and Application (Reviewer 1): Please expand the discussion on the potential applications of your model beyond cultivar resistance (e.g., field monitoring, ecological research) and address its limitations regarding color morphs, webbing, and damaged individuals.

Model Justification and Analysis (Reviewer 3): While not mandatory to add new models, a more robust justification for selecting YOLOv11 is needed. Please also include a deeper analysis of model failures based on the confusion matrix and discuss the implications of false positives/negatives.

Clarity and Presentation (All Reviewers): Enhance the manuscript's visual presentation by adding more figures, such as sample images, process visualizations, and improved data plots. A discussion on the cost-effectiveness and scalability of your approach, as raised by Reviewer 1, would also be valuable.

Introduction and Methods (Reviewer 2): Please revise the introduction as suggested and ensure all necessary methodological details, such as control treatments, are clearly mentioned.

When you submit your revision, please provide a detailed point-by-point response to all comments, explaining how you have addressed each one. Changes made in the manuscript should be clearly highlighted.

We look forward to receiving your revised manuscript.

Sincerely,

Reviewers' comments:

Reviewer's Responses to Questions

**Comments to the Author**

1. Is the manuscript technically sound, and do the data support the conclusions?

Reviewer #1: Yes

Reviewer #2: Yes

Reviewer #3: Partly

2. Has the statistical analysis been performed appropriately and rigorously?

Reviewer #1: Yes

Reviewer #2: Yes

Reviewer #3: Yes

3. Have the authors made all data underlying the findings in their manuscript fully available?

Reviewer #1: Yes

Reviewer #2: Yes

Reviewer #3: Yes

4. Is the manuscript presented in an intelligible fashion and written in standard English?

Reviewer #1: Yes

Reviewer #2: Yes

Reviewer #3: Yes

5. Review Comments to the Author

Reviewer #1: Dear Editor and Authors

The application of machine learning in species identification is a valuable and timely research direction. Overall, the system described here appears to have considerable potential beyond the development of resistant cultivars.

I do not see major issues with the model itself or with the way verification and validation were carried out. Tetranychus urticae life stages are generally distinct, except at the later nymphal stage, which the authors have appropriately acknowledged. My comments are as follows:

1. Several species within the genus Tetranychus are morphologically very similar, distinguishable only under high magnification. Would this model be useful for identifying other spider mite species? Furthermore, could variation in colour morphs (e.g. the red form of T. urticae) affect model accuracy?

2. Machine learning has become increasingly important in arthropod identification. For very small organisms such as T. urticae, the required magnification and resolution of the imaging system are critical. Could the authors clarify what level of image quality is necessary for the model to perform reliably? Dense webbing produced by spider mites may also obscure individuals and potentially interfere with classification—how is this accounted for?

3. In terms of cost-effectiveness for counting mites, how practical is this approach compared with existing methods? Some discussion of feasibility and scalability would strengthen the manuscript.

4. While developing resistant cultivars is one pest management strategy, it is not the only one. Could the authors discuss whether this model could also be applied in other scenarios, such as field monitoring, ecological research, or broader pest management programmes? For example, would individuals that have been partially consumed by predators such as Phytoseiulus persimilis be impossible to identify under the current model?

5. I encourage the authors to place stronger emphasis on the modelling aspects, highlighting the broader benefits of this approach. For instance, Lines 11–47 focus heavily on cultivar resistance, which is not the central subject of this study. Since the primary objective concerns the machine learning models, more space should be devoted to explaining how these models work, their principles, and their contributions to species identification and counting.

Final remark:

From my perspective, extending the model beyond life-stage identification to interspecific recognition would significantly enhance its utility for monitoring and management practices.

Reviewer #2: The manuscript is about an important subject and must be published. I have two comments, dealing with revising the introduction section and to mention the control treatments. The results of the manuscript will be important for different bioassay experiments and will help in phenotyping of spider mites.

Reviewer #3: The authors' study presents a novel approach to Automated detection and quantification of two-spotted spider mite life stages using computer vision for high-throughput in vitro assays. The study addresses a critical challenge in agriculture: the timely and accurate identification of damaging factors to optimize crop management and reduce reliance on chemical pesticides. They have clearly articulated their hypothesis, establishing a strong foundation for further exploration. The research was conducted with diligence, resulting in findings that invite discussion. However, some aspects should be addressed to enhance the scientific quality of the paper. By doing so, the rigor, clarity, and evaluation of the research will be significantly improved.

- The study only used YOLOv11 without benchmarking against other state-of-the-art CNNs (e.g., ResNet, EfficientNet, Vision Transformers). It is recommended to compare performance with at least 2–3 other architectures to validate Inceptionv3’s superiority.

- Based on the confusion matrix, analyze cases where the model fails. Discuss implications of false negatives (e.g., undetected infestations leading to crop loss).

- Relying on just two figures for this paper is inadequate. Enhancements to sample images and data visualization are necessary. Additionally, incorporating process visualization can enhance the overall quality of the paper, along with a greater use of images.

- Implementing data augmentation could enhance the model's performance by virtually increasing the amount of available data.

6. PLOS authors have the option to publish the peer review history of their article (what does this mean?). If published, this will include your full peer review and any attached files.

Reviewer #1: No

Reviewer #2: No

Reviewer #3: **Yes: **Abbas Ali Zamani

---

## [Author Response · Author response to Decision Letter 1]

25 Nov 2025

Please see the Word document for our point-by-point response to the reviewers' comments to the first review. Please see the second Word document for our point-by-point response to the formatting changing request following the second review.

---

## [Decision Letter · Decision Letter 1]

8 Dec 2025

Automated detection and quantification of two-spotted spider mite life stages using computer vision for high-throughput in vitro assays

PONE-D-25-49646R1

Dear Dr. David H. Gent and Co-authors,

On behalf of the editorial team of PLOS ONE, I am pleased to inform you that your revised manuscript, titled "Automated detection and quantification of two-spotted spider mite life stages using computer vision for high-throughput in vitro assays", has been accepted for publication.

The reviewers have evaluated your revisions and have confirmed that all previously raised concerns have been satisfactorily addressed. Both reviewers recommend acceptance of the manuscript in its current form.

We find that the study presents a novel and well-validated computer vision approach for automating the detection and quantification of Tetranychus urticae life stages. The work is methodologically sound, the results are clearly presented, and the manuscript makes a valuable contribution to the fields of entomology, agricultural science, and automated phenotyping.

Reviewers' comments:

Reviewer's Responses to Questions

**Comments to the Author**

1. If the authors have adequately addressed your comments raised in a previous round of review and you feel that this manuscript is now acceptable for publication, you may indicate that here to bypass the “Comments to the Author” section, enter your conflict of interest statement in the “Confidential to Editor” section, and submit your "Accept" recommendation.

Reviewer #1: All comments have been addressed

Reviewer #2: All comments have been addressed

2. Is the manuscript technically sound, and do the data support the conclusions?

Reviewer #1: Yes

Reviewer #2: Yes

3. Has the statistical analysis been performed appropriately and rigorously?

Reviewer #1: (No Response)

Reviewer #2: Yes

4. Have the authors made all data underlying the findings in their manuscript fully available?

Reviewer #1: Yes

Reviewer #2: Yes

5. Is the manuscript presented in an intelligible fashion and written in standard English?

Reviewer #1: Yes

Reviewer #2: Yes

6. Review Comments to the Author

Reviewer #1: All comments have been addressed, and I am satisfied for the manuscript to proceed to publication in PLOS ONE.

Reviewer #2: The manuscript has been considerably revised. It will add value to the acarological research. I accept it for publication.

7. PLOS authors have the option to publish the peer review history of their article (what does this mean?). If published, this will include your full peer review and any attached files.

Reviewer #1: No

Reviewer #2: No

---

## [Editor Report · Acceptance letter]

PONE-D-25-49646R1

PLOS One

Dear Dr. Gent,

I'm pleased to inform you that your manuscript has been deemed suitable for publication in PLOS One. Congratulations! Your manuscript is now being handed over to our production team.

Kind regards,

on behalf of

Dr. Muhammad Asif Qayyoum

Academic Editor

PLOS One